# Inducing skyrmions in ultrathin Fe films by hydrogen exposure

Pin-Jui Hsu[1,2], Levente Rózsa [1,3], Aurore Finco[1], Lorenz Schmidt[1], Krisztián Palotás [4,5], Elena Vedmedenko[1], László Udvardi[6,7], László Szunyogh[6,7], André Kubetzka[1], Kirsten von Bergmann [1] & Roland Wiesendanger[1]

Magnetic skyrmions are localized nanometer-sized spin configurations with particle-like properties, which are envisioned to be used as bits in next-generation information technology. An essential step toward future skyrmion-based applications is to engineer key magnetic parameters for developing and stabilizing individual magnetic skyrmions. Here we demonstrate the tuning of the non-collinear magnetic state of an Fe double layer on an Ir(111) substrate by loading the sample with atomic hydrogen. By using spin-polarized scanning tunneling microscopy, we discover that the hydrogenated system supports the formation of skyrmions in external magnetic fields, while the pristine Fe double layer does not. Based on ab initio calculations, we attribute this effect to the tuning of the Heisenberg exchange and the Dzyaloshinsky–Moriya interactions due to hydrogenation. In addition to interface engineering, hydrogenation of thin magnetic films offers a unique pathway to design and optimize the skyrmionic states in low-dimensional magnetic materials.

[1] Department of Physics, University of Hamburg, 20355 Hamburg, Germany. [2] Department of Physics, National Tsing Hua University, 30013 Hsinchu, Taiwan. [3] Institute for Solid State Physics and Optics, Wigner Research Centre for Physics, Hungarian Academy of Sciences, Budapest 1525, Hungary. [4] Institute of Physics, Slovak Academy of Sciences, 84511 Bratislava, Slovakia. [5] MTA-SZTE Reaction Kinetics and Surface Chemistry Research Group, University of Szeged, Szeged 6720, Hungary. [6] Department of Theoretical Physics, Budapest University of Technology and Economics, Budapest 1111, Hungary. [7] MTA-BME Condensed Matter Research Group, Budapest University of Technology and Economics, Budapest 1111, Hungary. Correspondence and requests for materials should be addressed to P.-J.H. (email: pinjuihsu@phys.nthu.edu.tw) or to L.Róz. (email: rozsa.levente@physnet.uni-hamburg.de)

Magnetic skyrmions are whirling spin textures displaying topological properties[1–4]. They offer promising perspectives for the future development of information technology based on their nanoscale size[5], robustness against defects[6], motion driven by low current densities[7], and the skyrmion Hall effect[8]. In order to engineer key features of the particle-like magnetic skyrmions, it is essential to tune the magnetic parameters of the thin films as well as their interfaces. In the presence of strong spin–orbit coupling, the Dzyaloshinsky–Moriya interaction[9,10] (DMI) emerges at surfaces and interfaces, which favors the formation of non-collinear spin states such as skyrmions. The magnetic ground state of the film is usually determined by the balance of the DMI energy, the magnetocrystalline anisotropy energy, the Zeeman energy, and the symmetric Heisenberg exchange interaction energy. The Heisenberg exchange can also contribute to the formation of skyrmions if ferromagnetic nearest-neighbor and antiferromagnetic next-nearest-neighbor interactions are taken into account[11].

In the quest for a knowledge-based material design for future skyrmionic devices, different routes are pursued. Up to date, the most widespread method for tuning magnetic parameters is interface engineering based on the growth of multilayered heterostructures. For example, by adding a Pd overlayer on top of Fe/Ir(111), the emergence of individual nanoscale magnetic skyrmions could be demonstrated by low-temperature spin-polarized scanning tunneling microscopy[5,12] (SP-STM). Moreover, sandwiching Co layers between different types of heavy metals led to the observation of magnetic skyrmions at room temperature[13–15], which has been explained by an enhancement of the DMI due to the multiple interfaces involved.

The role of hydrogen in the material embrittlement in metals and alloys stimulated detailed investigations on its molecular kinetics of adsorption, dissociation, and diffusion on and below metal surfaces[16,17]. The incorporation of hydrogen can significantly influence the magnetic properties of materials[18], for example by changing the effective magnetic moment, modifying the magnetic anisotropy[19–21], or tuning the interlayer exchange coupling[22]. However, very little is known about the role of hydrogen in tailoring non-collinear spin states, in particular, inducing magnetic skyrmions in ultrathin films and multilayers by means of hydrogen exposure has not been reported yet.

In the present study, we first demonstrate the tuning of the magnetic properties of the zero-field ground state in an Fe double layer on an Ir(111) single crystal substrate by loading the sample with atomic hydrogen. We observe two structural phases with distinct magnetic states. One of these hydrogen-induced phases (H1-Fe) is characterized by a spin spiral ground state with an increased magnetic period, by about a factor of three, compared to the spin spiral state of the pristine Fe double layer[23] (Fe-DL). The other hydrogen-induced phase (H2-Fe) is found to be ferromagnetic. With applied magnetic field, a transition from the hydrogen-induced spin spiral state to a magnetic skyrmion state is demonstrated experimentally and reproduced by simulations based on Heisenberg exchange interaction and DMI parameters determined from ab initio calculations. Remarkably, such a transition does not occur for the pristine Fe-DL, at least up to a magnetic field of 9 T. This drastic effect of hydrogen on non-collinear spin structures opens a further avenue for tuning the microscopic magnetic interactions responsible for creating and stabilizing individual skyrmions in ultrathin films as required for the future development of skyrmion-based devices.

## Results

**Hydrogen-induced structural phases.** Our earlier SP-STM studies of ultrathin Fe films on Ir(111) have demonstrated that the

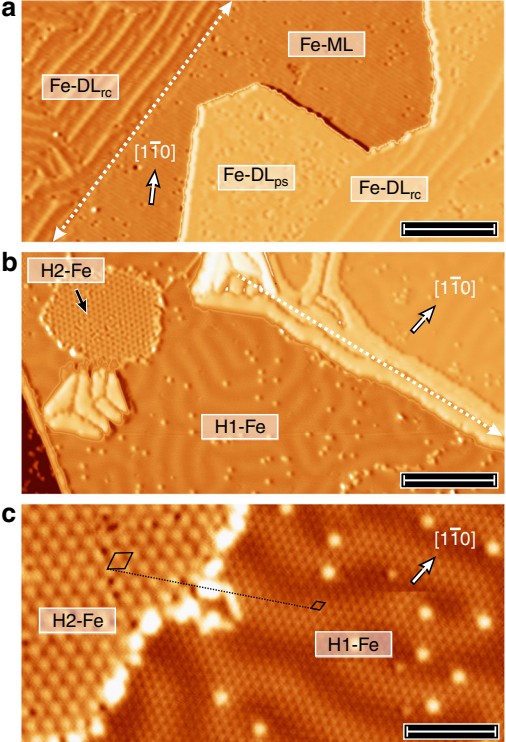

**Fig. 1** The pristine and the hydrogenated Fe double layer on Ir(111). **a** STM constant-current topography image of a sample with about 1.6 atomic layers of Fe grown on Ir(111). The dashed white line indicates a buried Ir step. The Fe-ML grows pseudomorphically, while the Fe-DL pseudomorphic areas (ps) coexist with a reconstruction (rc) formed by dislocation lines along the $[11\bar{2}]$ direction. Scale bar 15 nm. **b** STM constant-current topography image after hydrogen exposure (4.8 L) of a sample of about 2.3 atomic layers of Fe on Ir(111). The dashed white line indicates a buried Ir step. Two different H-induced superstructures on the Fe-DL are visible. Scale bar 15 nm. **c** Magnified view of the two hydrogen-induced superstructures on the Fe-DL. While the H1-Fe phase forms a p(2 × 2) superstructure, the H2-Fe phase has a larger unit cell and is rotated with respect to the high-symmetry directions. Scale bar 5 nm. Measurement parameters: $U = +0.2$ V for **a** and $U = -0.2$ V for **b**, **c**. $I = 1$ nA; $T = 4–5$ K; $B = 0$ T; W tip in **a** and Cr bulk tip in **b**, **c**. The images are partially differentiated and the contrast levels have been adjusted separately for different terraces

Fe monolayer (Fe-ML) grows pseudomorphically and exhibits a nanoskyrmion lattice with a magnetic period of about one nanometer[24]. When more Fe is deposited onto the Ir substrate, the tensile strain increases and the Fe-DL locally relieves the strain by the incorporation of dislocation lines[23]. Figure 1a shows a constant-current STM topography image of a sample with about 1.6 atomic layers of Fe on Ir(111), where pseudomorphic (Fe-DL$_{ps}$) and reconstructed (Fe-DL$_{rc}$) double-layer Fe areas coexist. The magnetic ground state of the pseudomorphic Fe-DL is a spin spiral state with a period of about 1.2 nm, which is stable in external magnetic fields of up to 9 T[23]. The morphology of the Fe-DL changes upon hydrogen exposure and subsequent annealing: as seen in Fig. 1b, the number of dislocation lines in the Fe-DL is significantly decreased and two different commensurate superstructures emerge, see high-resolution image in Fig. 1c. On the right side of Fig. 1c, a hexagonal lattice with twice the lattice constant of the substrate is observed, i.e., a p(2 × 2) superstructure with respect to the Ir(111) surface, which we assign to a hydrogen-loaded Fe-DL. Note that the modulation on the several nm scale in Fig. 1c is due to the magnetic structure of the film and will be discussed together with Fig. 2. On

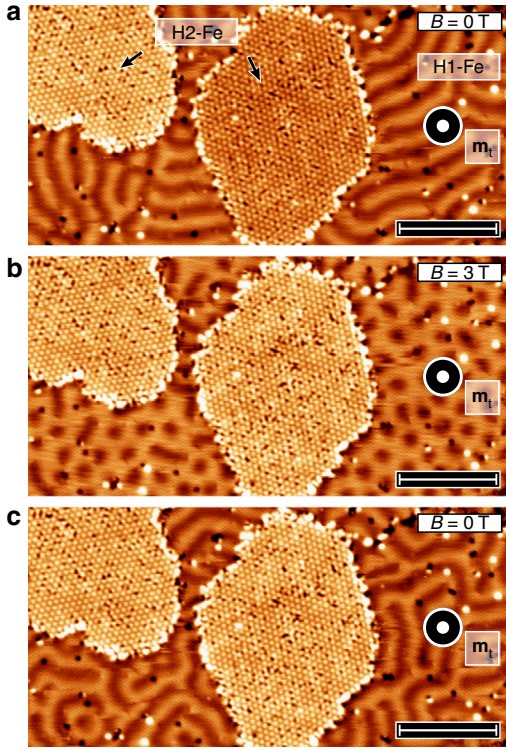

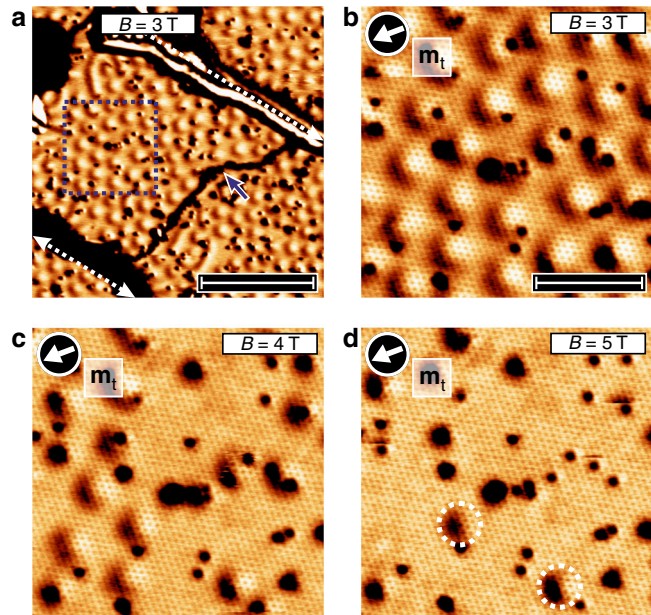

**Fig. 2** SP-STM images of the two different H-Fe double-layer areas at different applied magnetic fields. **a** SP-STM constant-current image of the H-induced superstructures on the Fe-DL without external magnetic field B. The apex of the Cr bulk tip has its magnetic moment $\mathbf{m_t}$ perpendicular to the surface, thus the tip is sensitive to the out-of-plane magnetization components of the sample. The periodic modulation of the contrasts for H1-Fe demonstrates spin spiral order, whereas the two-level contrast on H2-Fe reflects its ferromagnetic ground state. **b** Same as **a** but in a perpendicular magnetic field of $B = 3$ T. The dark dots and the surrounding brighter area in H1-Fe are magnetized in opposite out-of-plane directions. Both islands of H2-Fe now show the same magnetic contrast, i.e., they are magnetized along the same direction. **c** Same as **a**, also in zero-magnetic field. In this remanent state, the spin spiral in the H1-Fe area is more disordered and the H2-Fe islands remain in their field-polarized states. Measurement parameters: $U = -0.7$ V; $I = 1$ nA; $T = 4.2$ K. Scale bars 15 nm

**Fig. 3** Magnetic skyrmions in the H1-Fe phase. **a** SP-STM map of differential tunneling conductance of the hydrogenated Fe-DL, with large H1-Fe areas on three adjacent terraces. The Cr bulk tip used for this measurement is sensitive to the in-plane components of the sample magnetization; the dashed white lines are buried Ir steps and the black line indicated by the blue arrow is a phase domain between adjacent H1-Fe areas on the same terrace. It is evident that all magnetic objects show the identical spin-dependent two-lobe contrast, which demonstrates that they are skyrmions with unique rotational sense. Scale bar 25 nm. **b-d** Magnified views of the magnetic skyrmions at increasing external magnetic fields as indicated in the panels. Scale bar 8 nm. In **d**, two remaining skyrmions are shown by circles. For corresponding STM topography images, see Supplementary Fig. 4. Measurement parameters: $U = -0.2$ V; $I = 1$ nA; $T = 4.2$ K

the left side of Fig. 1c, a slightly rotated hexagonal superstructure with a larger lattice constant of 0.98 nm is observed, which is caused by a different hydrogen-induced phase of the Fe-DL. In the following, we refer to these two hydrogen-induced phases of the Fe-DL as H1-Fe and H2-Fe—see Supplementary Note 1 and Supplementary Figs. 1–3 for more details including atomic defects and possible structural models. Whereas these two phases might simply arise from different hydrogen concentrations, their sharp transition regions and a much higher stability of H2-Fe against tip-induced changes indicate different vertical positions of the hydrogen atoms with respect to the surface.

**Stabilization of individual magnetic skyrmions**. To investigate the magnetic state of the hydrogenated Fe-DL phases, we performed SP-STM measurements, which are sensitive to the projection of the local sample magnetization onto the quantization axis defined by the spin orientation of the probe tip[25]. Consequently, the measured spin-polarized tunnel current depends on the relative alignment of tip and sample magnetization, and the magnetic texture is reflected in constant-current SP-STM images such as the one in Fig. 2a. In the H1-Fe area, a modulation of magnetic origin can be observed with a period of about 3.5 nm

due to the presence of a spin spiral state, but the p($2 \times 2$) atomic superstructure is not resolved at this scale. Two separate areas of the H2-Fe can be seen in Fig. 2a. They exhibit a relative height difference of about 6 pm. Upon application of an external magnetic field, see Fig. 2b, the two H2-Fe areas exhibit the same height. We conclude that the H2-Fe is ferromagnetic and that the two areas in Fig. 2a have opposite magnetization directions. Due to the application of a magnetic field (Fig. 2b), the magnetization of the right H2-Fe island is switched. In this external magnetic field of 3 T, the spin spiral phase of the H1-Fe with a period of 3.5 nm is transformed into dots, which here represent the out-of-plane magnetization component opposite to the applied magnetic field. When the magnetic field is removed (Fig. 1c), the magnetization direction of the H2-Fe areas remains unchanged, whereas the magnetic dots of the H1-Fe structure transform back into a spin spiral phase, which differs only in details compared to the previous zero-magnetic-field state in Fig. 2a.

For examining the spin texture of the H1-Fe in more detail, we used a spin-polarized tip that is sensitive to the in-plane components of the sample magnetization. The magnetic objects emerging at 3 T with a diameter of about 3 nm are now imaged as two-lobe structures, and the brighter signal is always on the same side, see Fig. 3a and the magnified view in Fig. 3b. This is a signature of magnetic skyrmions with unique rotational sense[5]. When the external magnetic field is increased, see Fig. 3b–d, the field-polarized state is favored and the number of magnetic skyrmions decreases. In addition, due to the associated Zeeman energy, also the size of the individual magnetic skyrmions shrinks. Note that the features that remain unchanged by the

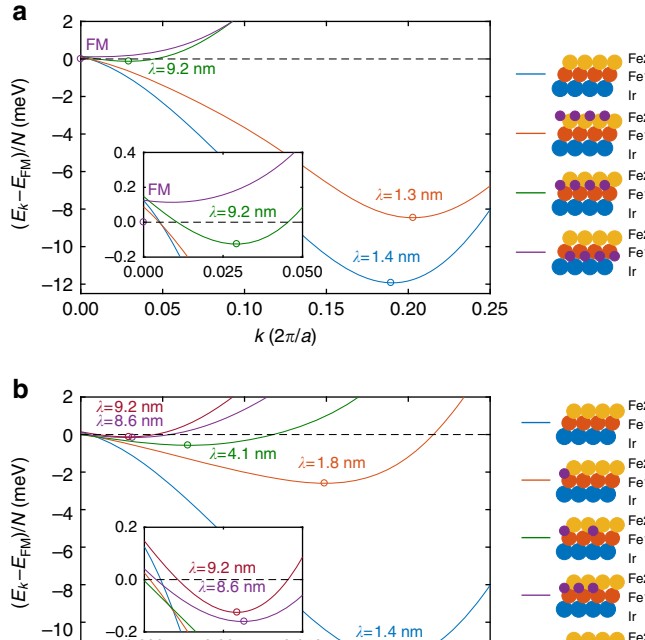

**Fig. 4** Influence of the H adsorption site and H concentration on the magnetic ground state from ab initio calculations. Energies of harmonic cycloidal spin spiral states per spin as a function of wave vector along the $[11\bar{2}]$ direction compared to the ferromagnetic state. **a** Dependence on adsorption site. From top to bottom in the legend: Fe double layer, H above the Fe layers in fcc hollow sites, H between the Fe layers in octahedral sites, and H between Ir and Fe in octahedral sites. **b** Dependence on H concentration. The concentration increases in steps of 0.25 ML from top to bottom in the legend. Circles denote the location of the minima converted to the magnetic period length $\lambda = 2\pi/k$, with the lattice constant a = 2.71 Å. The energy of the spiral at zero wave vector differs from that of the ferromagnetic state due to the magnetic anisotropy energy

external magnetic field are due to defects, see also Supplementary Note 2 and Supplementary Fig. 4.

**Ab initio calculations**. To understand the role of hydrogen for the different magnetic states in the Fe-DL, density-functional theory calculations were performed. Both the position and the concentration of H atoms were varied, for more details, see the Methods section, Supplementary Notes 3 and 4 and Supplementary Tables 1–4. The results for H layers in different vertical positions with respect to the surface are summarized in Fig. 4a, displaying the energy of the magnetic system as a function of spin spiral wave vector $k$. The ferromagnetic state is located at $k = 0$, and as $k$ is increased the angle between neighboring spins in the lattice also increases. For the pristine Fe-DL, which continues the fcc stacking of the Ir(111), we obtain a spin spiral ground state with a period of 1.4 nm (Fig. 4a), close to the experimentally observed value of 1.2 nm for pseudomorphic growth. In agreement with previous results for ultrathin Fe films on the Ir(111) surface[12,24], we find a right-handed cycloidal rotation of the spin spirals originating from interfacial DMI. When a full layer of H is adsorbed above the Fe layers in fcc hollow sites, the magnetic period of the spin spiral ground state decreases slightly to 1.3 nm, i.e., in this configuration, the H does not considerably alter the magnetic ground state. In comparison, a significant increase of the magnetic period to 9.2 nm is observed for a layer of H adsorbed in octahedral sites between the Fe atomic layers. Finally,

the calculations show that the ground state is ferromagnetic when the H is placed in octahedral sites at the Fe–Ir interface.

These drastic differences in the magnetic ordering may be attributed to the strong dependence of the microscopic magnetic interactions on the hybridization between the atoms, including direct hopping processes between the Fe atoms and indirect processes mediated by the Ir substrate or the H atoms. As demonstrated in Fig. 4, the hybridization sensitively depends on the vertical and horizontal adsorption site of hydrogen. However, since the overlap of the wave functions is directly connected to the distance between the atoms, we find a strong correlation between the interlayer distances and the period of the obtained magnetic ground state. It has been demonstrated in earlier publications[12,24,26–28] that the strong hybridization between the Ir and Fe atoms is responsible for the formation of non-collinear ground states with short magnetic periods in similar systems, and that a decreased interlayer distance leads to a reduction in the period length. If the H is adsorbed between the two Fe layers, it increases their distance by about 30 pm compared to the pristine Fe-DL, while the adsorption at the Fe–Ir interface increases both the Ir–Fe1 and Fe1–Fe2 distances by about 20 pm. In comparison, if the H is adsorbed above the Fe layers, it only has a minimal effect on the interlayer distances (cf. refs. [18,29]) and the magnetic structure. For more details on the interlayer distances, see Supplementary Note 3 and Supplementary Tables 1–3.

Based on these results, we can explain the experimentally observed ferromagnetic H2-Fe phase by the incorporation of H at the Fe–Ir interface in octahedral sites, see calculated ferromagnetic ground state in Fig. 4a. In contrast, the H1-Fe phase exhibits a spin spiral ground state with a period of ~3.5 nm. Comparison with the calculations suggests that this phase might be due to H atoms being located between the two Fe layers, but with an intermediate H concentration between the H-free Fe-DL with a spin spiral period of 1.4 nm and the complete H layer with 9.2 nm, see Fig. 4a. This assignment is in agreement with the experimental indications that the hydrogen in the two different H-Fe phases is located in different vertical positions with respect to the surface.

In the theoretical calculations, we increased the concentration of H atoms between the Fe layers in steps of 0.25 ML, and indeed found a gradual enhancement of the magnetic period from 1.4 to 9.2 nm as displayed in Fig. 4b. At the same time, the distance between the Fe layers increases, meaning that the outer Fe2 layer becomes less hybridized with the inner Fe1 layer and the Ir substrate as discussed above. Our calculations reveal that in this geometry the hybridization between the Fe and H orbitals also contributes to the increase of the magnetic period independently of the layer separation—for a more detailed discussion, see Supplementary Note 5 and Supplementary Fig. 5.

**Spin model calculations**. Our results show that both a change of the vertical position of H atoms as well as a variation of the H concentration can lead to modified magnetic properties of the hydrogenated Fe-DL. In the following, we look in more detail into those two systems, which represent best the experimentally observed H1-Fe and H2-Fe phases, i.e. the one with 0.50 ML of H between the Fe layers and the one where the H is located at the Fe–Ir interface, corresponding to the spin spiral ground state with 4.1 nm period in Fig. 4b and the FM ground state in Fig. 4a. To obtain a deeper understanding of the microscopic mechanisms, we derive effective interaction parameters based on the spin spiral dispersion relations (Fig. 4), described by the Hamiltonian

$$H = -\frac{1}{2}\sum_{\langle i,j\rangle_1} J_1 \mathbf{S}_i \mathbf{S}_j - \frac{1}{2}\sum_{\langle i,j\rangle_2} J_2 \mathbf{S}_i \mathbf{S}_j - \frac{1}{2}\sum_{\langle i,j\rangle_1} \mathbf{D}_{ij}(\mathbf{S}_i \times \mathbf{S}_j) - \sum_i K(S_i^z)^2 - \sum_i \mu_s B S_i^z.$$

(1)

**Table 1 Parameters used for the spin dynamics simulations**

| System | Fe-DL | H1-Fe | H2-Fe |
|---|---|---|---|
| $J_1$ (meV) | 33.18 | 51.00 | 65.14 |
| $J_2$ (meV) | −17.14 | −17.00 | −19.72 |
| $D$ (meV) | −1.36 | −1.09 | −0.27 |
| $K$ (meV) | 0.25 | −0.01 | 0.25 |
| $\mu_s$ ($\mu_B$) | 2.89 | 2.84 | 2.58 |
| $f = (J_1 + 3J_2)/J_1$ | −0.55 | 0.00 | 0.09 |

Spin model parameters used for describing the low-energy behavior of the magnetic systems considered in the ab initio calculations—see Eq. (1). The Fe-DL, H1-Fe, and H2-Fe parameter sets describe the ab initio calculations for the pristine Fe-DL (top curve in the legend of Fig. 4a), 0.50 ML H in octahedral positions between the Fe layers (middle curve in the legend of Fig. 4b) and H in octahedral positions between Ir and Fe (bottom curve in the legend of Fig. 4a), respectively. Negative values of $f = (J_1 + 3J_2)/J_1$ mean that the Heisenberg exchange interactions prefer a non-collinear ground state. Positive and negative values for the atomic Heisenberg exchange interactions denote ferromagnetic and antiferromagnetic couplings, respectively. The negative sign of the DMI indicates a right-handed rotation of the spin spiral. For the anisotropy, positive and negative values describe easy-axis and easy-plane types, respectively

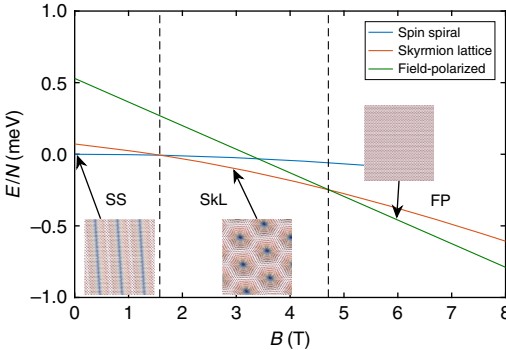

**Fig. 5** Results of the spin dynamics simulations. Zero-temperature phase diagram of the model describing the H1-Fe system with the parameters from Table 1, displaying the energy per spin in different magnetic structures as a function of the strength of the external field $B$ applied perpendicular to the surface. Dashed black lines indicate the transition fields between the spin spiral, skyrmion lattice, and field-polarized states. Insets display the ground states observed at selected field values

The system is modeled by classical spins $S_i$ occupying atomic positions of a single-layer hexagonal lattice, with nearest- and next-nearest-neighbor Heisenberg exchange interactions $J_1$, $J_2$, nearest-neighbor DMI $|D_{ij}| = D$, on-site anisotropy $K$, atomic magnetic moment $\mu_s$, and the external field $B$ applied perpendicularly to the surface. Analogously to a micromagnetic description, the model given by Eq. (1) is expected to correctly describe the system's properties only at long wavelengths and low energies—see Supplementary Note 6 and Supplementary Figs. 6 and 7 for more details.

The derived parameters are summarized in Table 1 for the Fe-DL, the H1-Fe, and H2-Fe structures. Within the description of Eq. (1), two mechanisms may prefer a non-collinear order of the spins, namely the DMI and the frustration of the symmetric Heisenberg exchange interactions[11], the latter of which is encapsulated in the dimensionless parameter $f = (J_1 + 3J_2)/J_1$, where negative values indicate a spin spiral ground state—see Supplementary Note 6 and Supplementary Table 5 for details. For the pristine Fe-DL, the calculations yield $f < 0$, meaning that frustration plays an indispensable role in the formation of the spin spiral ground state; the rather large DMI of −1.36 meV (or −6.99 mJ m$^{−2}$) also favors non-collinear magnetic order and leads to a right-handed unique rotational sense. In contrast, in the H1-Fe, the frustration of symmetric Heisenberg exchange interactions is reduced ($f \approx 0$) due to the increase of $J_1$; here, the

non-collinear magnetic ground state can form only because of the significant value of the DMI (−1.09 meV or −6.05 mJ m$^{−2}$). The ferromagnetic ground state for the H2-Fe arises because $f > 0$, and at the same time, the DMI is nearly quenched (−0.27 meV or −1.46 mJ m$^{−2}$) and cannot compete with the collinearity enforced by the large $J_1$ and the out-of-plane anisotropy. We attribute this pronounced decrease in the DMI to the significant increase of the Fe–Ir interlayer distance in the H2-Fe phase, which presumably reduces the hopping processes between the magnetic layer and the substrate exhibiting high spin–orbit coupling, regarded as the main mechanism for the emergence of interfacial DMI[10,30]. Finally, Fig. 5 displays the low-temperature phase diagram of the H1-Fe structure obtained from spin dynamics simulations, with illustrations of the resulting spin spiral, skyrmion lattice, and field-polarized ground states. The skyrmion lattice phase is the ground state in a regime from about 1.6 to 4.8 T, in good agreement with the magnetic field regime where magnetic skyrmions are observed experimentally (Fig. 3). For the pristine Fe-DL, the simulations agree with the experiments in determining a spin spiral ground state up to field values of 9 T.

## Discussion

As shown in Fig. 2, both the H1-Fe and the H2-Fe structures display two kinds of point defects appearing as bright protrusions and dark pits in the STM constant-current topography images. According to a series of growth studies in Supplementary Note 1 and Supplementary Figs. 1 and 2, repeated dosages of hydrogen on the sample lead to a decrease in the number of bright protrusions and an increase in the number of dark pits, which makes it possible to identify the former as hydrogen vacancies and the latter as additional hydrogen atoms. Note that the defects can have a reversed contrast in differential tunneling conductance images as shown in Fig. 3 and described in detail in Supplementary Note 2 and Supplementary Fig. 4.

As demonstrated in ref. [31] for the Pd/Fe/Ir(111) system, surface defects act as local perturbations on the potential landscape, exhibiting pinning effects on non-collinear magnetic spin textures. Figures 1–3 display that in the H1-Fe structure, both types of defects tend to prefer in-plane magnetization directions, being primarily located off the centers of skyrmions (Fig. 3) or at the boundary between the up and down magnetized areas of spin spirals as imaged by an out-of-plane magnetized tip (Fig. 2a, c). Vacancies and additional adatoms may also be thought of as local modifications of the hydrogen concentration between or above the Fe layers, which change the magnetic interactions between the Fe atoms—see also Supplementary Note 5 and Supplementary Fig. 5.

However, our measurements indicate that the atomic-scale defects only have a minor influence on the nanometer-scale spin spiral period and the size of magnetic skyrmions. Furthermore, we always found that the pristine Fe-DL is in the spin spiral state and the H2-Fe structure is in the ferromagnetic state, regardless of the presence of dislocation lines, defects, or the value of the external magnetic field up to 9 T. Magnetic skyrmions can only be stabilized in the H1-Fe structure, and although the field value necessary for collapsing them into the field-polarized state slightly depends on the local defects[31], skyrmions can typically be observed between 3 and 5 T. This means that at a fixed external field value, the hydrogen-induced magnetic state is stable, and can only be structurally transformed into a disordered phase, for example, by an intentional increase of hydrogen dosage or extreme tunneling conditions as discussed in Supplementary Note 1 and Supplementary Fig. 2. We expect that the preparation of a well-defined and extended H1-Fe phase becomes possible after optimizing the amount of hydrogen and the annealing temperature as illustrated by the growth studies in Supplementary Note 1 and Supplementary Fig. 1.

In conclusion, we demonstrated how the magnetic parameters and the corresponding magnetic period length in ultrathin Fe films may be tuned by hydrogen exposure. SP-STM measurements performed on an Fe-DL on Ir(111) revealed two hydrogenated phases with different atomic structures. While the H2-Fe structure is ferromagnetic, the H1-Fe structure displays a spin spiral ground state at zero field and skyrmions with a unique rotational sense in the presence of an applied field. Ab initio calculations suggest that the incorporation of H into different vertical positions and in different concentrations leads to a modification of the frustrated Heisenberg exchange interactions and DMI responsible for the formation of non-collinear magnetic order. This can result either in a ferromagnetic ground state or in an increase of the spin spiral period and the appearance of skyrmions in applied magnetic fields.

Most recent propositions for tailoring the magnetic interactions for skyrmion applications have focused on the appropriate choice of the magnetic layer or the heavy metal materials with strong spin–orbit coupling responsible for the appearance of interfacial DMI[12,15,32]. As demonstrated in this paper, the adsorption of non-magnetic elements with low atomic numbers such as hydrogen can also result in a significant modulation of the interactions directly by hybridization with the magnetic atoms and indirectly by modifying the atomic structure, which influences the hybridization between the magnetic layer and the substrate. This phenomenon, which is expected to be ubiquitous for impurities of low-Z elements[33], can potentially be turned into a new platform for the controlled design and development of skyrmionic materials.

## Methods

**Sample preparation and characterization**. A clean surface of the Ir(111) substrate was prepared in ultra-high vacuum with a base pressure of $\leq 1 \times 10^{-10}$ mbar. Cycles of $Ar^+$ ion sputtering with an emission current of 25 mA and a beam energy of 800 eV were performed at an Ar gas pressure of about $8 \times 10^{-5}$ mbar. A sample current of about 10 μA was detected during the $Ar^+$ ion sputtering. Besides that, several cycles of oxygen annealing of Ir(111) was carried out by varying the substrate temperature from room temperature to 1350 °C under oxygen exposure. The oxygen pressure was reduced stepwise from $5 \times 10^{-6}$ mbar to $5 \times 10^{-8}$ mbar.

After the Ir(111) substrate was clean, about 1.5–2.5 ML of Fe was evaporated with a deposition rate of about 0.5 ML per min using molecular beam epitaxy. These films were subsequently exposed to atomic hydrogen obtained by cracking hydrogen gas molecules at high temperature in a dedicated hydrogen source. An amount of 4.8 L of atomic hydrogen was dosed onto the Fe-DL/Ir(111) at room temperature, followed by a post-annealing treatment at about 600 K for 10 min, resulting in extended areas of hydrogenated Fe. The effect of the amount of hydrogen and the post-annealing treatment is discussed in more detail in Supplementary Note 1 and Supplementary Figs. 1 and 2.

**SP-STM measurements**. SP-STM measurements were performed by using chemically etched bulk Cr tips in a home-built low-temperature STM setup cooled down to 4.2 K by liquid helium. For topographic images, the STM was operated in the constant-current mode with the bias voltage ($U$) applied to the sample. Differential tunneling conductance (d$I$/d$U$) maps were acquired by lock-in technique with a small voltage modulation $U_{mod}$ (10–30 mV) added to the bias voltage and a modulation frequency of 5–6 kHz. External magnetic fields of up to 9 T were applied to the sample along the out-of-plane direction.

For bulk Cr tips, the magnetization direction of the tip apex atom determines the quantization axis of the spin sensitivity. Because this direction is not known a priori, we derive it from the measured images and symmetry considerations. The reconstructed Fe-DL displays three rotational domains of spin spirals with 120° angle between the wave vector directions in the different domains[23]. Both in-plane and out-of-plane magnetized tips are sensitive to the spiral modulation. However, while all rotational domains of spin spirals have the same corrugation amplitude with an out-of-plane sensitive tip, they are imaged with different amplitudes when the tip is sensitive to in-plane components[23]. For example, when the in-plane tip magnetization is parallel to the propagation direction of a cycloidal spin spiral, one observes maximum magnetic contrast, whereas the amplitude in the other two rotational domains is reduced by a factor of cos(±120°). Magnetic skyrmions imaged with a spin-polarized tip sensitive to the out-of-plane component appear axially symmetric. When an in-plane sensitive tip is used, the magnetic contribution to the signal vanishes for the center of the skyrmion and its surrounding, because there tip and sample magnetizations are orthogonal. Instead the maximum magnetic contrast is observed on the in-plane parts of the skyrmion that are parallel or antiparallel to the tip magnetization, and a two-lobe structure appears. When the two-lobe structures of all skyrmions appear the same, they have the same rotational sense due to the interfacial

DMI[5,34]. Magnetic skyrmions stabilized by interfacial DMI are typically cycloidal due to the symmetry selection rules[6,35,36].

**Density-functional theory calculations**. Geometry optimizations were carried out using the Vienna Ab initio Simulation Package[37–39] (VASP). The Fe double layer was modeled by 7 Ir and 2 Fe atomic layers in fcc growth within a $1 \times 1$ unit cell, with about 18 Å empty space vertically between the slabs. We used $a = 2.71$ Å for the in-plane lattice constant of the Ir(111) surface. Different H adsorption sites were investigated by adding a H layer to the above slab. Varying the H concentration was performed by using a $2 \times 2$ unit cell and 1, 2 or 3 H atoms between the Fe layers, corresponding to 0.25, 0.50 and 0.75 ML coverages, respectively. We used pseudopotentials from the projector-augmented wave method with the Perdew–Wang 91 (PW91) parametrization[40] of the generalized gradient approximation, and a $15 \times 15 \times 1$ Monkhorst–Pack **k**-mesh. The vertical positions of the top Ir, the two Fe and the H layers were allowed to relax until the forces became smaller than 0.01 eV Å$^{-1}$. We determined the Bader charges belonging to the atoms using the method from refs. [41–43].

The magnetic structure was studied by using the fully relativistic Screened Korringa–Kohn–Rostoker (SKKR) method[44,45] within the atomic sphere approximation. For the pristine Fe-DL, we used 10 Ir and 2 Fe layers, as well as 3 layers of empty spheres (vacuum) between semi-infinite bulk Ir and semi-infinite vacuum in a $1 \times 1$ unit cell. For the hydrogenated system, we used 9 Ir, 2 Fe, 1 H, and 3 vacuum layers. The interlayer distances corresponded to the values optimized by VASP, and the radii of the atomic spheres were determined in a way that minimizes overlap, with the Bader charges computed from VASP serving as a benchmark against which the charges within the atomic spheres were compared. Partial H coverages were considered by applying the coherent potential approximation in a $1 \times 1$ unit cell, corresponding to a random occupation of the available adsorption sites with the probability determined by the H concentration. The vertical positions corresponded to the averaged value over atoms belonging to the same layer from VASP. For the self-consistent calculations, the Vosko–Wilk–Nusair parametrization[46] for the exchange–correlation potential was used within the local spin density approximation. We applied the relativistic torque method[47] to determine the magnetocrystalline anisotropy energy, as well as pairwise Heisenberg exchanges and DMIs between the Fe atoms within a radius of $8a$, corresponding to 240 intralayer and 225 interlayer neighbors for each spin. From the spin model parameters, we calculated the energy per spin in a harmonic cycloidal spin spiral configuration,

$$\mathbf{S}_i = \mathbf{n} \cos \mathbf{k} \mathbf{R}_i \pm \hat{\mathbf{k}} \sin \mathbf{k} \mathbf{R}_i, \qquad (2)$$

where **n** is the outwards pointing normal vector of the surface, **k** is the in-plane wave vector with direction $\hat{\mathbf{k}}$, and the ± signs describe right- and left-handed rotational senses, respectively. The simplified model parameters displayed in Table 1 were obtained by calculating the spin spiral dispersion relation from Eq. (1) and fitting it to the dispersion relation determined based on ab initio calculations at the center of the Brillouin zone encompassing the minimum position. The self-consistent calculations also yielded the spin magnetic moments, the value of which was averaged over the two Fe layers to obtain $\mu_s$.

**Spin dynamics simulations**. The field dependence of the magnetic ground state was studied on a $128 \times 128$ hexagonal lattice with periodic boundary conditions using the simplified model parameters discussed above. For the H1-Fe system, three different spin configurations were considered: a cycloidal spin spiral state with wave vector minimizing the energy in the absence of external field, a skyrmion lattice corresponding to a triple-**k** state constructed from the above wave vector, and the collinear field-polarized state. The energy was minimized at each magnetic field value by the numerical solution of the stochastic Landau–Lifshitz–Gilbert equation[48]. The choice of Gilbert damping coefficient $\alpha = 1$ ensured fast convergence, and the energy differences between the final states of the simulations and the corresponding local energy minima were ~$10^{-6}$ meV per atom. The boundary conditions did not allow for a change in the wave vector during the simulations. The wave vector of the minimum energy state only weakly depends on the magnetic field, apart from a short range close to the transition between the skyrmion lattice and the field-polarized states.

**Data availability**. The data supporting the findings of this study are available from the corresponding authors upon request.

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

## Acknowledgements

We thank J. Hagemeister, N. Romming, M. Dupé, B. Dupé, and S. Heinze for insightful discussions. Financial support was provided by the Deutsche Forschungsgemeinschaft via SFB 668, by the European Union via the Horizon 2020 research and innovation program under grant agreement no. 665095 (MAGicSky), by the Alexander von Humboldt Foundation, by the National Research, Development and Innovation Office of Hungary under project nos. K115575 and FK124100, by the Slovak Academy of Sciences via the SASPRO Fellowship (project no. 1239/02/01), by the Tempus Foundation via the Hungarian State Eötvös Fellowship and by the MOST under project no. 107-2112-M007-001-MY3, Taiwan.

## Author contributions

P.-J.H. performed the experiments. P.-J.H., A.F., L.S., A.K., and K.v.B. analyzed the data. L.R. and K.P. performed the VASP calculations. L.R., K.P., L.U., and L.S. performed the SKKR calculations and analyzed the results. L.R. and E.V. performed the spin dynamics simulations and discussed the data. P.-J.H. and L.R. prepared the figures, P.-J.H., K.v.B., L.R., and R.W. wrote the manuscript. All authors discussed the results and provided inputs to the manuscript.

## Additional information

**Competing interests:** The authors declare no competing interests.

