## [Peer Review File · Nature Communications]

Reviewers' comments:

Reviewer #1 (Remarks to the Author):

See attached file.

Reviewer #2 (Remarks to the Author):

In this work, Hsu et al. investigate the alteration of magnetic skyrmions at the Ir(111)/Fe(2ML) surface after exposure to hydrogen. They observe two structural phases and report the observation of magnetic skyrmions in one of them only. They interpret their results using Density Functional Theory calculations accounting for intercalation of H atoms in the Fe bilayer. Following the authors' interpretation, the nanoskyrmions (~1.4nm) obtained in the pristine case are enlarged (~9 nm) when H atoms intercalate between the two Fe monolayers. This interpretation seems reasonable and is well supported by theory. Micro magnetic modeling is performed to extract the system parameters. This is an overall beautiful and consistent study that demonstrates the impact of hydrogenation on the presence of skyrmions.

That being said, I am not convinced that this study constitutes a breakthrough that deserves publication in Nature Communication, for two main reasons:

(i) I don't see how the present work advances the understanding of chiral magnetism at the surface of transition metals. Explaining the skyrmion radius enhancement due to H-intercalation is reasonable but not unexpected. Considering that this enhancement is essentially due to lattice distortion (increasing distance between the two Fe layers), such distortions are not surprising.

(ii) I am not convinced the present work could open new avenues for the engineering of skyrmion lattices. Upon H exposure, both H1-Fe and H2-Fe phases appear in a non-controlled manner, which renders the skyrmion engineering using this technique quite questionable. In addition, the presence of skyrmion lattice seems to be very sensitive to structural distortions, omnipresent in ultrathin films, which makes it hardly adapted to engineering purposes.

In summary, this is a nice study performed in a rigorous manner; I also acknowledge that this is probably the first observation of skyrmion modification under H exposure. That being said, I do not see how disruptive this observation is and how it can open "novel avenues" for skyrmion

engineering. In my view, this manuscript deserves publication in a specialized journal, but not in a broad-audience journal such as Nature Communication.

Reviewer #3 (Remarks to the Author):

In their manuscript "Inducing skyrmions in ultrathin Fe films by hydrogen exposure", Dr. Hsu and co-workers describe spin-STM experiments and a theoretical interpretation of significant effects of hydrogen absorption on the magnetic properties of two monolayer thick Fe/Ir(111) films.

Magnetic skyrmions are currently a hot topic and a substantial community is interested in materials and properties. This paper is timely, and the finding that hydrogen absorption allows stabilization of magnetic skyrmions in films that do not support skyrmion formation in absence of hydrogen (at least not at the available field strength) is interesting to the community. The STM experimental data looks convincing and the theoretical interpretation seems reasonable. The paper is written well and it will be interesting to the audience of nature communications, I recommend publication.

Reviewer #1 (Remarks to the Author):

Review of "Inducing skyrmions in ultrathin Fe films by hydrogen exposure" by Dr. Hsu et al. submitted to *Nature Communications*.

The manuscript describes the influence of hydrogen exposure on the magnetic spin texture in an Fe double layer (Fe-DL) on an Ir(111) single crystal substrate. Using STM topography, they revealed two kinds of superstructures on the Fe-DL. One is termed as H1-Fe while the other is termed as H2-Fe. Using SP-STM technique, they found that H1-Fe is a spin spiral ground state at zero magnetic field, while H2-Fe is ferromagnetic. With applied magnetic field, they demonstrated a transition from the spin spiral state to a magnetic skyrmion state. Furthermore, they tried to reproduce their experimental results by simulations based on Heisenberg exchange interaction and DMI parameters determined from *ab initio* calculations. Their calculations suggested that the incorporation of H into different vertical positions and in different concentrations leads to the two different magnetic states.

As far as I know, the result (hydrogen-induced skyrmion state on a surface) is novel and their experimental technique (SP-STM) is well established. Of course, the control of the microscopic magnetic interactions responsible for creating and stabilizing individual skyrmions in ultrathin films is an important topic for the future development of skyrmion-based devices. Basically, I think the manuscript is appropriate for publication on *Nature Communications*. I should point out, however, a concern to be solved before acceptance of the manuscript.

In Figure 1b&c, a lot of defects appear to be introduced in the hydrogenated Fe double layer. Different kinds of defects are observed for H1-Fe (protrusions) and H2-Fe (pits). Since the magnetic spin texture of a thin film specimen is likely to be affected by such defects, the authors must describe the nature of the two kinds of defects and discuss their influence clearly.

In addition, from a viewpoint of practical applications, it would be helpful for readers to comment on the stability of the hydrogen-induced magnetic state. Furthermore, it appears strange that Discussion contains only concluding sentences. So, I suggest to the authors including the two topics - the nature and influence of defects and the stability of the hydrogenated magnetic state - in Discussion before concluding sentences.

Minor revision(s)

(1) In Supplementary Note 2: Page 3, Line 11: " temperature und subsequent" should be "temperature and subsequent".

End of Review

We thank the Reviewers for their careful evaluation of our manuscript as well as for the useful comments. Our replies are typeset in blue. The changes in the manuscript are summarized at the end of the reply.

(I) The reply and corresponding changes to the questions of Reviewer #1:

Comment (1): *“The manuscript describes the influence of hydrogen exposure on the magnetic spin texture in an Fe double layer (Fe-DL) on an Ir(111) single crystal substrate. Using STM topography, they revealed two kinds of superstructures on the Fe-DL. One is termed as H1-Fe while the other is termed as H2-Fe. Using SP-STM technique, they found that H1-Fe is a spin spiral ground state at zero magnetic field, while H2-Fe is ferromagnetic. With applied magnetic field, they demonstrated a transition from the spin spiral state to a magnetic skyrmion state. Furthermore, they tried to reproduce their experimental results by simulations based on Heisenberg exchange interaction and DMI parameters determined from ab initio calculations. Their calculations suggested that the incorporation of H into different vertical positions and in different concentrations leads to the two different magnetic states.”*

“As far as I know, the result (hydrogen-induced skyrmion state on a surface) is novel and their experimental technique (SP-STM) is well established. Of course, the control of the microscopic magnetic interactions responsible for creating and stabilizing individual skyrmions in ultrathin films is an important topic for the future development of skyrmion-based devices. Basically, I think the manuscript is appropriate for publication on Nature Communications. I should point out, however, a concern to be solved before acceptance of the manuscript.”

Reply: We would like to thank the Reviewer for identifying the essence of our work. We have replied to all comments in a point-by-point fashion below.

Comment (2): *“In Figure 1b&c, a lot of defects appear to be introduced in the hydrogenated Fe double layer. Different kinds of defects are observed for H1-Fe (protrusions) and H2-Fe (pits). Since the magnetic spin texture of a thin film specimen is likely to be affected by such defects, the authors must describe the nature of the two kinds of defects and discuss their influence clearly.”*

“In addition, from a viewpoint of practical applications, it would be helpful for readers to comment on the stability of the hydrogen-induced magnetic state. Furthermore, it appears strange that Discussion contains only concluding sentences. So, I suggest to the authors including the two topics - the nature and influence of defects and the stability of the hydrogenated magnetic state - in Discussion before concluding sentences.”

Reply: We thank the Reviewer for this comment. Following his/her suggestion, we expanded the Discussion section by the paragraphs about the role of defects and the stability of the magnetic structure. We also expanded Supplementary Note 2 with the identification of different types of defects, since we found that this is connected to repeated dosages of hydrogen. See items 1, 4, 5, 10, 11, 12, 13 in the list of changes.

Regarding the defects, while Fig. 1c only displays bright protrusions in the H1-Fe and dark pits in the H2-Fe structures, we found that actually both types of defects can appear in both structures in constant-current topography images as shown in, e.g., Fig. 2. In the series of growth studies in Supplementary Note 2, the number of bright

protrusions on top of the H1-Fe and at the edge of the H2-Fe in Supplementary Fig. 2f is obviously reduced after exposure to an additional amount of 4.8 L hydrogen as shown in Supplementary Fig. 2i, which allows us to identify the nature of these protrusions as hydrogen vacancies. Further increasing the amount of hydrogen leads to an increase in the number of dark pits according to Supplementary Figs. 3a and 3b, indicating that they correspond to extra hydrogen atoms. The hydrogen vacancies (bright protrusions) on top of the H1-Fe exhibit the same geometrical shape, indicated by white triangles pointing along the [11-2] direction in Supplementary Fig. 4a. Therefore, we are able to infer the positions for which the H atoms are possibly missing in the structural models, marked by solid red circles in Supplementary Figs. 4b and c.

We agree with the Reviewer that the defects or vacancies can influence the non-collinear magnetic spin textures; for example, the pinning effect by native in-layer defects and artificially built Co clusters on spin spirals and magnetic skyrmions in the Pd/Fe/Ir(111) system was recently investigated in C. Hanneken *et al.*, *New J. Phys.* 18, 055009 (2016). We find that in the present H1-Fe structure both types of defects tend to be located off the centres of magnetic skyrmions, similarly to the observations in the previous work cited above. Given that the potential energy landscape can be locally modified via defects, magnetic skyrmions are stabilized at higher magnetic fields and presumably their preferred positions can be located in the vicinity of the pinning sites. Such pinning effects also enable the distortions of magnetic skyrmions and spin spirals as demonstrated in the study by C. Hanneken *et al.* These observations are relevant to possible influences of defects on the surface of hydrogenated Fe-DL, which is now accessible in the revised manuscript.

However, we found that the defects do not significantly influence the period of spin spirals and the size of skyrmions, or the fact that the latter can only be observed in the H1-Fe structure among the systems studied in the manuscript, in the 3-5 T field range. This indicates that the magnetic phases are robust against local modifications, and can only be influenced significantly together with the atomic structure by forming different hydrogen-induced phases.

Comment (3): “(1) In Supplementary Note 2: Page 3, Line 11: " temperature und subsequent" should be "temperature and subsequent".”

Reply: We thank the Reviewer for pointing out this typo, which we have corrected in the resubmitted version, see item 8 in the list of changes.

(II) The reply and corresponding changes to the questions of Reviewer #2:

Comment (1): “In this work, Hsu *et al.* investigate the alteration of magnetic skyrmions at the Ir(111)/Fe(2ML) surface after exposure to hydrogen. They observe two structural phases and report the observation of magnetic skyrmions in one of them only. They interpret their results using Density Functional Theory calculations accounting for intercalation of H atoms in the Fe bilayer. Following the authors' interpretation, the nanoskyrmions (~1.4nm) obtained in the pristine case are enlarged (~9 nm) when H atoms intercalate between the two Fe monolayers. This interpretation seems reasonable and is well supported by theory. Micro magnetic modeling is performed to extract the system parameters. This is an overall beautiful and consistent study that demonstrates the impact of hydrogenation on the presence of

skyrmions.

That being said, I am not convinced that this study constitutes a breakthrough that deserves publication in *Nature Communication*, for two main reasons:

(i) I don't see how the present work advances the understanding of chiral magnetism at the surface of transition metals. Explaining the skyrmion radius enhancement due to H-intercalation is reasonable but not unexpected. Considering that this enhancement is essentially due to lattice distortion (increasing distance between the two Fe layers), such distortions are not surprising.”

Reply: We thank the Reviewer for his/her conclusion that “*This is an overall beautiful and consistent study that demonstrates the impact of hydrogenation on the presence of skyrmions.*” However, we feel that the Reviewer did not fully appreciate the importance of our work, and we would like to clarify the significant advancements as described in our manuscript by emphasizing the following points:

1. What we report on is not a “*skyrmion radius enhancement*” as judged by Reviewer 2. As it is clearly stated in the abstract and repeated on page 2 in the last paragraph of the Introduction, “the hydrogenated system supports the formation of skyrmions in external magnetic fields, while the pristine Fe double layer does not”, which indicates a fundamental difference between the two systems. The ground state of the Fe-DL system is a spin spiral state with approximately 1.2 nm period, which is not influenced by an external magnetic field up to 9 T. The H1-Fe structure displays a spin spiral ground state with 3.5 nm period, and this is the only system studied in the manuscript where the formation of isolated skyrmions is reported. Concerning the sizes of magnetic skyrmions, as shown in Fig. 3, the average diameter of the skyrmions is around 3 nm at 3 T, which decreases upon increasing the applied field. On the other hand, the H2-Fe structure supports the ferromagnetic state. These results clearly demonstrate that hydrogenation of the pristine Fe-DL leads not only to a modulation of the magnetic structure, but magnetic phase transitions as well.

To further emphasize this point in the manuscript, we expanded the Discussion section by explaining that the Fe-DL stays in the spin spiral state, while the H1-Fe structure transforms into the skyrmion state under external field. We also added a sentence about the field dependence of the Fe-DL based on spin dynamics simulations at the end of the theoretical section. See items 3, 5 in the list of changes.

2. We consider it very surprising that Reviewer 2 finds the enhancement of the magnetic period due to the adsorption of H “*not unexpected*”. First, according to our knowledge H adsorption effects on non-collinear magnetism has hardly been investigated in the past, especially from the experimental side, as mentioned in the 2nd paragraph on page 2. Based on the summarizing paragraph of the Reviewer’s report, the Reviewer also shares this view. Second, our *ab initio* calculations indicate that the magnetic ground state of the system depends very sensitively on the adsorption site and concentration of hydrogen due to the influence of hybridization between the Fe layer and the Ir substrate as well as due to the direct hybridization between Fe and H, as explained in the main text and discussed in Supplementary Notes 3-5 in more detail. Figure 4 demonstrates that the effects range from a slight decrease in the magnetic period to the formation of a ferromagnetic state. From the experimental side, two structures with different magnetic properties are reported. Taking all of these facts into account, we find it very difficult to judge what would be the “*expected*” behaviour of the system when exposed to hydrogen in the absence of a combined rigorous experimental and theoretical investigation.

3. In our opinion, the fact that “*distortions are not surprising*” does not mean that the presented hydrogen effect is trivial. It is known that the microscopic magnetic interactions delicately depend on the electronic structure of the material, leading to a wide variety of types of magnetic orderings observable in nature. Although it is expected that hydrogen adsorption influences the electronic structure, its effect on the magnetic ordering seems to be very difficult to predict. As an example, Supplementary Fig. 5b demonstrates that tuning the hydrogen concentration while keeping the interlayer distances fixed also leads to a modulation of the spin spiral period, also mentioned in the 4th paragraph on page 4 of the main text. However, since interatomic hybridization processes directly depend on the distances between the atoms, despite these complications it is possible to observe a strong correlation between the period of the magnetic ground state and the interlayer distances as discussed in the manuscript, which leads to a relatively simple but qualitatively correct explanation of the effects. Although several earlier theoretical investigations have pointed out the importance of lattice relaxations in determining the correct magnetic structure (see references [12,24,26,27,28] in the 2nd paragraph on page 2), tuning the interlayer distance in the experiments is generally not possible and has not been investigated in ultrathin films in detail in this context. As mentioned in the 2nd paragraph of the Introduction and the last paragraph of the Discussion, most experimental efforts have focused on the choice of transition metal-heavy metal interfaces (see references [13-15]), instead of the effect of the light element hydrogen on the non-collinear magnetic state as in the present study.

In order to further clarify the role of the electronic structure in the formation of the magnetic ground state, of which the interlayer distances are only a single but certainly important aspect, we extended the theoretical discussion in the second paragraph on page 2, see item 2 in the list of changes. We also comment on the importance of hybridization effects in the last paragraph of the main text, already present in the previous version.

4. We think that another reason why our work “*advances the understanding of chiral magnetism*” is the role played by frustrated exchange interactions in the formation of non-collinear ground states, see Table 1. A large number of previous investigations on skyrmions, including the overwhelming majority of the experimental studies, is based on a micromagnetic description only containing ferromagnetic exchange interaction, Dzyaloshinsky–Moriya interaction and magnetocrystalline anisotropy, where only the Dzyaloshinsky–Moriya interaction may favour the formation of skyrmions. This way, our work once again establishes a connection between recent theoretical investigations discussing frustrated exchange interactions (see e.g., reference [11] and the references therein) and experimental observations.

To emphasize this point, we expanded the second paragraph on page 6 of the Discussion section, see item 5 in the list of changes.

Comment (2): “(ii) I am not convinced the present work could open new avenues for the engineering of skyrmion lattices. Upon H exposure, both H1-Fe and H2-Fe phases appear in a non-controlled manner, which renders the skyrmion engineering using this technique quite questionable. In addition, the presence of skyrmion lattice seems to be very sensitive to structural distortions, omnipresent in ultrathin films, which makes it hardly adapted to engineering purposes.”

Reply: We cannot agree that the hydrogenated phases appear in a completely “*non-controlled manner*”. This question is discussed in detail in Supplementary Note 2. In a

series of systematic growth studies, it is demonstrated in Supplementary Fig. 2 that repeated H adsorption and post-annealing cycles lead to the appearance of well-defined, extended and connected H1-Fe areas where skyrmions can be observed, because the H2-Fe areas coalesce.

It is true that the appearance of magnetic skyrmions is linked to the structural changes induced by the hydrogen adsorption; however, Fig. 3 clearly demonstrates the shape and size of skyrmions inside the H1-Fe structure is completely well-defined, despite the presence of local defects. Furthermore, we do not agree that structural distortions and defects themselves may prohibit the engineering of non-collinear spin structures, given the significant number of observations of magnetic skyrmions even in amorphous systems such as in references [8,13,14,15]. Therefore, we believe that hydrogen-induced structural distortions are not a limiting factor, but an alternative route for developing and stabilizing individual magnetic skyrmions in ultrathin magnetic films.

We expanded the Discussion section on pages 5-6 to emphasize the stability of the observed non-collinear spin structures despite the presence of defects and structural distortions. In Supplementary Note 2 (page 3), we added a sentence explaining how the H2-Fe and consequently the H1-Fe areas coalesce during repeated annealing cycles. See items 5, 9 in the list of changes.

Comment (3): *“In summary, this is a nice study performed in a rigorous manner; I also acknowledge that this is probably the first observation of skyrmion modification under H exposure. That being said, I do not see how disruptive this observation is and how it can open “novel avenues” for skyrmion engineering. In my view, this manuscript deserves publication in a specialized journal, but not in a broad-audience journal such as Nature Communication.”*

Reply: As already discussed above, we would like to emphasize that the key outcome of our studies is not simply a “skyrmion modification under H exposure” as commented by the Reviewer. On the contrary, the major finding of our work is that adsorbing a low-Z element such as atomic hydrogen on an ultrathin magnetic film can strongly modify non-collinear magnetic order, even resulting in the emergence of skyrmions which are absent in the pristine system without hydrogen adsorption. We thus expect that this effect should stimulate further investigations well beyond the specific material choice of H, Fe and Ir as discussed in the present manuscript. We believe that the presentation of the results is appropriate for a broader readership, while the content is in agreement with the mission statement “Papers published by the journal represent important advances of significance to specialists within each field”, thereby meeting the publication criteria of Nature Communications.

(III) The reply and corresponding changes to the questions of Reviewer #3:

Comment (1): *“In their manuscript “Inducing skyrmions in ultrathin Fe films by hydrogen exposure”, Dr. Hsu and co-workers describe spin-STM experiments and a theoretical interpretation of significant effects of hydrogen absorption on the magnetic properties of two monolayer thick Fe/Ir(111) films. Magnetic skyrmions are currently a hot topic and a substantial community is interested in materials and properties. This paper is timely, and the finding that hydrogen absorption allows stabilization of magnetic skyrmions in films that do not support skyrmion formation in absence of hydrogen (at least not at the available field*

strength) is interesting to the community. The STM experimental data looks convincing and the theoretical interpretation seems reasonable. The paper is written well and it will be interesting to the audience of nature communications, I recommend publication.”

Reply: We thank the Reviewer for his/her statements and appreciate the positive evaluation of our work.

(IV) List of changes (following the numbering of the resubmitted version)

1. **Page 2, Results, 1st paragraph, sentence** beginning with “In the following, we refer to these [...] including possible structural models.” have been changed to “In the following, we refer to these [...] including atomic defects and possible structural models.” in response to Comment (2) of Reviewer #1.
2. **Page 4, Results, 2nd paragraph** beginning with “These drastic differences in the magnetic ordering [...]” was expanded in response to Comment (1) of Reviewer #2.
3. **Page 5, Results, 2nd paragraph, sentence** beginning with “For the Fe-DL, [...]” was added in response to Comment (1) of Reviewer #2.
4. **Page 5, Discussion, 1st and 2nd paragraphs** beginning with “As shown in Fig. 2, [...]” and reference 31 were added in response to the Comment (2) of Reviewer #1.
5. **Page 6, Discussion, 1st paragraph** beginning with “However, the results indicate [...]” was added in response to the Comment (2) of Reviewer #1 and Comments (1)-(2) of Reviewer #2.
6. **Page 6, Discussion, 2nd paragraph** beginning with “*Ab initio* calculations suggested that [...]” was extended in response to the Comment (1) of Reviewer #2.
7. **Page 10, Acknowledgements** was updated along with the author contact information.
8. **Supplementary Note 2, page 3, 2nd paragraph, sentence** beginning with “Upon additional exposure of the same sample”, the typo pointed out in the Comment (3) of Reviewer #1 was corrected.
9. **Supplementary Note 2, page 3, 2nd paragraph, sentence** beginning with “However, since the H₂-Fe islands [...]” was rephrased in response to the Comment (2) of Reviewer #2.
10. **Supplementary Note 2, page 5, 1st paragraph** beginning with “The repeated dosages of hydrogen [...]” was added in response to the Comment (2) of Reviewer #1.
11. **Supplementary Note 2, page 6, 2nd paragraph, sentence** beginning with “Since the hydrogen vacancies (bright protrusions) [...]” was added in response to the Comment (2) of Reviewer #1.

12. **Supplementary Fig. 3, page 5**, the white and black dashed circles were added to panel a to indicate the atomic defects on the hydrogenated Fe-DL surface. The caption has been revised accordingly.

13. **Supplementary Fig. 4, page 6**, the white triangles were added to mark the hydrogen vacancies in panel a and solid red circles were added to infer possible missing H atoms from the proposed structural models in panels b and c. The caption has been revised accordingly.

Reviewers' comments:

Reviewer #1 (Remarks to the Author):

Reviewer 1's comments on NCOMMS-17-29387A "Inducing skyrmions in ultrathin Fe films by hydrogen exposure" by Dr. Hsu et al. submitted to Nature Communications.

I am almost satisfied with the authors' revised manuscript in full response to my previous suggestions. Now, I vote for a publication in Nature Communications after they respond to the following two suggestions.

(1) In connection with reviewer 2's comment (ii), I suggest to the authors that they add notes in the main text on their prospect to create pure H1-Fe phase without coexistence of H2-Fe phase to engineer skyrmion states in a fully controlled manner.

(2) In connection with the nature of the surface defects (hydrogen vacancy or extra hydrogen atoms on the surface), I also suggest to the authors that they prepare Figure 4c to evaluate the influence of the H concentration on the surface (above Fe2). The figure should be similar to Figure 4b evaluating the influence of the H concentration on the interlayer (between Fe1 and Fe2).

End of Reviewer 1's comments

Reviewer #2 (Remarks to the Author):

The authors have replied carefully and convincingly to my comments. I particularly appreciate that they took the time to develop their arguments and added appropriate discussion to the manuscript. I indeed missed the point of H-driven spin-spiral-to-skyrmion transition, and I recognize the stability of the process and the in-depth analysis provided in the paper. I therefore support publication in Nature Communications as it is.

Reviewer #3 (Remarks to the Author):

As I indicated in my earlier report, I believe that this paper will be interesting to the audience of nature communications. In this revised version the authors addressed all comments provided by all reviewers, I recommend publication of this manuscript.

We thank the Reviewers for their careful evaluation of our manuscript as well as for the positive recommendation. Our replies are typeset in blue. The changes in the manuscript are highlighted in blue and also summarized at the end of this reply.

(I) The reply and corresponding changes to the questions of Reviewer #1:

Comment (1): “*I am almost satisfied with the authors’ revised manuscript in full response to my previous suggestions. Now, I vote for a publication in Nature Communications after they respond to the following two suggestions.*”

(1) In connection with reviewer 2’s comment (ii), I suggest to the authors that they add notes in the main text on their prospect to create pure H1-Fe phase without coexistence of H2-Fe phase to engineer skyrmion states in a fully controlled manner. (2) In connection with the nature of the surface defects (hydrogen vacancy or extra hydrogen atoms on the surface), I also suggest to the authors that they prepare Figure 4c to evaluate the influence of the H concentration on the surface (above Fe2). The figure should be similar to Figure 4b evaluating the influence of the H concentration on the interlayer (between Fe1 and Fe2).”

Reply: We thank the Reviewer for the positive evaluation and the useful suggestions. In response to Comment (1), we added a sentence about the preparation of a well-defined and extended H1-Fe phase to the Discussion section of the main text. Regarding Comment (2), it had already been shown in Figure 4a and mentioned in the corresponding discussion in the last paragraph on page 3 that adsorbing H in the fcc site above the Fe2 layer only minimally modifies the spin spiral period, from 1.4 nm to 1.3 nm. We found that this can be understood as a net effect of a slightly increased interlayer distance leading to a larger period and the hybridization preferring a smaller period for this adsorption site. To illustrate this, we added Supplementary Figure 5c displaying the requested concentration dependence for the H above the Fe layers, and extended Supplementary Note 5 correspondingly. We also added a sentence to the Discussion section about the connection between the concentration dependence and the defects, referring to these newly added data. Since this effect overall does not have a strong impact as compared to the adsorption of H between the Fe layers in the octahedral position and the corresponding decrease in the spiral period contradicts the experimental findings, we deemed that this figure would not be suitable for the main manuscript in the present context, but rather for Supplementary Figure 5c.

(II) The reply and corresponding changes to the questions of Reviewer #2:

Comment (1): “*The authors have replied carefully and convincingly to my comments. I particularly appreciate that they took the time to develop their arguments and added appropriate discussion to the manuscript. I indeed missed the point of H-driven spin-spiral-to-skyrmion transition, and I recognize the stability of the process and the in-depth analysis provided in the paper. I therefore support publication in Nature Communications as it is.*”

Reply: We thank the Reviewer for assessing the essence of our work and appreciate his/her positive comments.

(III) The reply and corresponding changes to the questions of Reviewer #3:

Comment (1): *“As I indicated in my earlier report, I believe that this paper will be interesting to the audience of nature communications. In this revised version the authors addressed all comments provided by all reviewers, I recommend publication of this manuscript.”*

Reply: **We thank the Reviewer again for his/her positive evaluation of our work.**

(IV) List of changes (following the numbering of the resubmitted version)

1. **Page 6, Discussion, 1st paragraph, sentence** beginning with “Vacancies and additional adatoms [...]” was added in response to the Comment (2) of Reviewer #1. The following paragraph was also slightly rephrased for clarity.
2. **Page 6, Discussion, 2nd paragraph, sentence** beginning with “We expect that [...]” was added in response to the Comment (1) of Reviewer #1.
3. **Supplementary Figure 5c** about the concentration dependence of the spin spiral period for the H adsorbed above the Fe layers was added in response to the Comment (2) of Reviewer #1. The caption and the discussion in the second paragraph of Supplementary Note 5 on page 10 were also modified accordingly.

Reviewers' Comments:

Reviewer #1 (Remarks to the Author):

I have read the authors' revised manuscript and additional Supplementary materials. Now that they have carefully addressed my previous two comments, I recommend a prompt publication of their manuscript on Nature Communications. Congratulations.

We thank the Reviewer #1 for his/her careful evaluation of our manuscript as well as for the positive recommendation. Our reply is typeset in blue. The changes in the manuscript are summarized at the end of this reply.

(I) The reply and corresponding changes to the questions of Reviewer #1:

Comment (1): *“I have read the authors' revised manuscript and additional Supplementary materials. Now that they have carefully addressed my previous two comments, I recommend a prompt publication of their manuscript on Nature Communications. Congratulations.”*

Reply: **We thank the Reviewer for his/her positive evaluation of our work.**

(IV) List of changes (following the numbering of the resubmitted version)

1. **Page 2, Introduction, 3rd paragraph, sentence** “...spin structures open a new avenue...” was changed to “...spin structures opens a further avenue...” in response to the editorial request.
2. **Page 3, Results, 1st paragraph, sentence** “...- see Supplementary Note 2 and supplementary Fig. 2-4...” was changed to “..- see Supplementary Note 1 and Supplementary Figs. 1-3...” in response to the editorial request. Same changes have been made throughout the main text and the Supplementary Information.
3. **Figure 1**, the figure has been cited in order and the label of scale bar has been moved into the caption, same for all figures in the main text and the Supplementary Information.
4. **Supplementary Notes 1 and 2 as well as Supplementary Figures 1-4 are now numbered in the order they appear in the main text.**
5. **Figure labels and equations have been updated** in response to the editorial request.